# Patient Preferences or Provider Pressure? The Relationship Between Coercive Contraceptive Care and Preferred Contraceptive Use

**DOI:** 10.3390/healthcare13020145

**Published:** 2025-01-14

**Authors:** Laura E. T. Swan, Lindsay M. Cannon, Madison Lands, Iris Huimeng Zhao

**Affiliations:** 1Reproductive Equity Action Lab, School of Medicine and Public Health, University of Wisconsin-Madison, Madison, WI 53706, USA; 2Reproductive Equity Action Lab, Department of Sociology, Center for Demography and Ecology, University of Wisconsin-Madison, Madison, WI 53706, USA; 3Collaborative for Reproductive Equity, Department of Obstetrics and Gynecology, University of Wisconsin-Madison, Madison, WI 53706, USA; 4Department of Sociology, Center for Demography and Ecology, University of Wisconsin-Madison, Madison, WI 53706, USA

**Keywords:** reproductive autonomy, healthcare provider bias, family planning care, patient-centered care, contraceptive counseling, preferred contraception

## Abstract

**Background/Objectives**: Coercion in contraceptive care occurs when healthcare providers unduly influence patients to use or not use birth control. Contraceptive coercion is antithetical to quality patient-centered care. However, it is unclear how experiencing contraceptive coercion relates to patients’ lives and contraceptive outcomes. In this study, we examined associations between contraceptive coercion and a patient-centered outcome: preferred contraceptive use. **Methods**: In 2023, we used the Prolific panel to recruit reproductive-aged people in the USA who were assigned female at birth. Our analytic sample included surveyed participants who had ever talked to a healthcare provider about contraception (*N* = 1197). We conducted chi-square and regression analyses to investigate associations between contraceptive coercion and preferred contraceptive use. We added context by mapping the current and preferred contraceptive method(s) for participants who experienced coercion and were not using their preferred method(s). **Results**: After adjusting for potential confounders, participants who reported downward coercion (pressure *to not* use birth control) at their last contraceptive counseling were less likely to be using their preferred contraceptive method(s). The odds of using preferred contraception did not differ significantly based on whether participants experienced upward contraceptive coercion (pressure *to use* birth control). Patterns in unmet contraceptive preference for patients experiencing coercion include use of the pill when it is not the desired method and unmet desire for permanent contraception. **Conclusions**: In this study, patients who perceived pressure from a provider to not use birth control were less likely to be using their preferred contraceptive method(s). Promoting reproductive autonomy requires comprehensive, patient-centered, and unbiased contraceptive care.

## 1. Introduction

It is increasingly clear that patients commonly experience coercion in their contraceptive care. Emerging evidence suggests that over a third of people in the USA who can become pregnant have experienced contraceptive coercion in their lifetime [1,2]. This healthcare provider-based contraceptive coercion occurs when providers unduly influence patients either *to use* (upward contraceptive coercion) or *to not use* (downward contraceptive coercion) birth control [1,3].

Contraceptive coercion has structural as well as interpersonal roots and manifestations. For example, public health policies and programs have led to a systematic push for highly effective methods of birth control (e.g., long-acting reversible contraceptives [LARCs] like intrauterine devices and implants) over less effective methods [4,5], even though patients often prioritize factors beyond effectiveness at preventing pregnancy [6]. Qualitative research reveals that many patients find this tiered effectiveness counseling to be biased and coercive, as they feel pressure from providers to use a LARC method when that was not their desire [4,5,7,8,9,10].

Research also documents cases in which patients are denied wanted forms of permanent contraception because of providers’ assumptions about patients’ future reproductive desires or due to guidelines meant to prevent sterilization abuse [11,12,13]. In either case, despite possible well-meaning intentions held by individual providers, policymakers, and healthcare systems, these forms of contraceptive coercion limit patients’ reproductive autonomy.

It is likely that patients’ contraceptive use varies based on the content and quality of their contraceptive counseling, but little research has explored how experiences of contraceptive coercion relate to patients’ lives and contraceptive outcomes. Of particular interest is how contraceptive coercion might impact patients’ ability to access and use their preferred method(s) of contraception or make an autonomous choice to not use contraception. A growing body of literature uses this metric of preferred contraceptive use as a key patient-centered outcome where patients themselves define success in contraceptive care rather than considering, for example, high levels of contraceptive uptake to be the primary outcome of interest [14,15,16,17].

There are several pathways through which coercive contraceptive care could be linked to a decreased likelihood of preferred contraceptive use. First, after being pressured by a provider to use a contraceptive method that they do not want to use, a patient may ultimately use the method either long-term or temporarily. A qualitative study provides empirical support for the latter circumstance, documenting that patients perceiving coercion in their contraceptive counseling may accept providers’ birth control suggestions in order to end the healthcare encounter; however, these methods are often rapidly discontinued [18]. Second, patients may be refused their preferred method(s) by a provider and subsequently be unable to access and use that/those method(s). Third, encountering coercive contraceptive care may cause some patients to perceive using a non-preferred contraceptive method as their only option; they may accept another method that the provider will give them or use another non-preferred method that they can access independently from the healthcare system (e.g., condoms). In each of these theoretical scenarios, patients’ experiences of contraceptive coercion present as an insurmountable barrier to preferred contraceptive use. To be sure, some patients are undoubtedly able to overcome coercion as a barrier and access their preferred contraception (or choose to not use contraception at all). However, it is unknown how commonly this occurs and what the relationship may be between contraceptive coercion and contraceptive behavior.

To date, only one study has quantitatively connected contraceptive coercion to preferred contraceptive use. This study, conducted in the Appalachian region of the United States, linked experiences of contraceptive coercion to a reduced likelihood of preferred contraceptive use [1]. However, this relationship has not been investigated more broadly, beyond the Appalachian region, which is a geographically and culturally unique area of the United States where healthcare experiences and outcomes may differ from those across the country. In the current study, we build on this existing research, using a diverse, national sample to investigate the relationship between contraceptive coercion and preferred contraceptive use.

## 2. Materials and Methods

### 2.1. Participants and Procedures

In the spring of 2023, we recruited respondents through Prolific, a diverse national panel with documented reliability and thousands of vetted participants [19]. Using Prolific’s prescreening features, we recruited reproductive-aged people (18–49 years old) living in the United States who were assigned female at birth, oversampling racial/ethnic, gender, and sexual minorities to establish a diverse sample. Prolific uses a variety of validity checks (e.g., identity verification, bot detection, attention checks) to vet participants and ensure rigorous data collection. This research was approved by the University of Wisconsin-Madison Institutional Review Board.

We collected our targeted 1500 online survey responses, compensating participants with $4 for each completed survey ($19/h based on the average response time of 12.32 min, which is above Prolific’s suggested pay rate). We excluded from analysis 100 respondents who did not meet study eligibility based on their responses to validity check items (*n* = 7 over age 49, *n* = 93 who were not assigned female at birth). We also excluded participants who had never talked to a healthcare provider about birth control (*n* = 201), as these individuals were not asked follow-up questions about contraceptive counseling. Finally, we excluded participants who were missing data on key study variables (*n* = 2), bringing our analytic sample to 1197.

### 2.2. Measures

Our online survey included open- and closed-ended questions about participants’ health and contraceptive care.

#### 2.2.1. Independent Variable: Contraceptive Coercion

We measured contraceptive coercion at the most recent contraceptive counseling with the five-item Coercion in Contraceptive Care Checklist, which dichotomously (0 = no, 1 = yes) measures the two dimensions of contraceptive coercion: upward and downward coercion.

Two items measured downward coercion (pressure to not use birth control): (1) “The healthcare provider would not give me the birth control method I wanted” and (2) “The healthcare provider made me feel that I should not use birth control”. Participants who answered affirmatively to either of these two items were coded as experiencing downward contraceptive coercion (=1).

Three items measured upward coercion (pressure to use birth control): (1) “The healthcare provider made me keep using a birth control method that I wanted to stop using”; (2) “The healthcare provider made me feel like I had to use birth control”; and (3) “The healthcare provider made me use a specific birth control method”. Participants who answered affirmatively to any of these three items were coded as experiencing upward contraceptive coercion (=1).

Additional details about the development and validation of this measure are published elsewhere [2].

#### 2.2.2. Dependent Variable: Preferred Contraceptive Use

We measured preferred contraceptive use dichotomously based on responses to several survey questions. First, we asked about respondents’ current contraceptive method(s), providing a list of 17 options, including an “other” write-in option and an option indicating that no method is being used. Participants could select multiple methods.

Then, we asked about participants’ hypothetical contraceptive use if they “did not have to worry about cost and could use any type of birth control methods available”. We asked “would you want to change your birth control use” for participants who had reported using one or more contraceptive method(s) and asked “would you want to begin using a method” for participants who had reported not currently using any contraceptive method. We coded those who chose “no” or “don’t know” as using their preferred method (=1) and those who chose “yes” as not using their preferred method (=0). This conceptualization of preferred contraceptive use is in line with existing research and recent calls for patient-centered outcome measures [1,14,15,16,17,20].

We also asked participants who indicated that they were not using their preferred contraceptive method(s) to identify what method(s) they would use if they did not have to worry about cost and could use any method available, providing the same list of 17 options used to assess current contraceptive use.

#### 2.2.3. Covariates

We controlled for a series of potential confounders that are theoretically related to both contraceptive coercion and use of preferred contraception [21]. Participants reported information about their age, education level, health insurance status, race/ethnicity, sexual orientation, marital status, and the amount of time since their last contraceptive counseling.

### 2.3. Data Analysis

First, we used univariate statistics to describe our study sample (*N* = 1197). Next, we conducted chi-square tests to assess the relationship between contraceptive coercion and preferred contraceptive use. Then, we used logistic regression to assess the association between contraceptive coercion and preferred contraceptive use adjusted for potential confounders. We ran separate regression models for any contraceptive coercion, upward coercion, and downward coercion. We conducted these analyses in Stata 18 [22]. Finally, we used R version 4.3.1 (packages: tidyverse, ggplot2, ggsankey) to produce Sankey diagrams that map current and preferred contraceptive method(s) for the subsample of participants who experienced contraceptive coercion and were not using their preferred contraceptive method(s) [23,24].

## 3. Results

### 3.1. Sample Characteristics

Table 1 shows the distribution of sociodemographic characteristics in our sample. The majority of participants were cisgender women (94.9%). The mean age was 32.78 years. Two-thirds of the sample had either attended some college (32.6%) or had a bachelor’s degree (37.1%). The majority of participants were insured (90.9%). The sample was racially and ethnically diverse with 65.4% identifying as non-Hispanic White, 15.2% as non-Hispanic Black, 8.4% as Hispanic, 6.9% as non-Hispanic Asian, and 4.0% as non-Hispanic mixed race. Most identified as heterosexual (65.1%). About two-thirds were not married (64.9%). Nearly half of participants (46.7%) had been counseled on contraception by a healthcare provider in the past year.

### 3.2. Coercion and Preferred Contraceptive Use

Table 2 shows the distribution of contraceptive coercion and use of preferred contraception in the sample. Over one in six participants (*n* = 221, 18.5%) reported experiencing contraceptive coercion during their last contraceptive counseling. Upward coercion (*n* = 183, 15.3%) was more commonly reported than downward coercion (*n* = 51, 4.3%). Participants who reported that they experienced any form of contraceptive coercion at their last contraceptive counseling were significantly less likely to be using their preferred method(s) of contraception than participants who did not experience contraceptive coercion (80.1% vs. 87.2% respectively, *X*^2^ (1) = 7.50, *p* = 0.006). In separate analyses based on the direction of contraceptive coercion, participants who experienced downward coercion were significantly less likely to report using their preferred method(s) of contraception than those without this experience (70.6% vs. 86.6% respectively, *X*^2^ (1) = 10.28, *p* = 0.001). Although patients who experienced upward coercion were less likely to report using their preferred method(s) of contraception than those who did not experience upward coercion, this difference was not statistically significant (81.4% vs. 86.7% respectively, *X*^2^ (1) = 3.54, *p* = 0.060).

Figure 1 shows the associations between contraceptive coercion at last contraceptive counseling and the use of preferred contraception from logistic regression models (Appendix A show coefficients for the full models, including sociodemographic controls). Both before and after accounting for controls, experiencing any contraceptive coercion at last counseling was associated with significantly lower odds of using one’s preferred method(s) of contraception (adjusted OR = 0.63, 95% CI = 0.42–0.94). This relationship was largely driven by downward coercion, as, on average, downward contraceptive coercion was significantly associated with lower odds of using one’s preferred contraceptive method(s) net of controls (adjusted OR = 0.39, 95% CI = 0.20–0.75). Upward contraceptive coercion was not significantly associated with preferred use of contraception in either adjusted or unadjusted models (adjusted OR = 0.73, 95% CI = 0.47–1.13). In sensitivity analyses limiting the sample to only those whose last contraceptive counseling was in the past year, the results remained substantively unchanged.

Of the 221 participants (18.5%) who experienced some form of contraceptive coercion, 44 participants were not using their preferred contraceptive method(s), representing a higher proportion than those who were not using their preferred contraceptive method(s) but did not experience coercion (20% vs. 13%). Figure 2 documents nuances in the current and preferred contraceptive method(s) for this subsample of study participants (*n* = 44) who both experienced contraceptive coercion at their last contraceptive counseling and were not currently using their preferred method(s) of contraception. Although this is a relatively small subsample, we provide this context as critical and novel information about unmet contraceptive needs for patients who have experienced coercion in their contraceptive care.

One pattern is evident in the figure: a relatively large proportion of these participants would like to be using permanent contraception but are not using that method. Specifically, almost half (*n* = 7) of the 15 participants who experienced downward coercion and are not using their preferred contraception indicated that they would prefer to use permanent contraception; instead, they were currently using no method (*n* = 2), barrier methods (*n* = 2), the pill (*n* = 2), or LARCs (*n* = 1). Among those who experienced upward coercion and are not using their preferred contraception (*n* = 34), about a third (*n* = 10) would prefer to use permanent contraception; instead, they were using no method (*n* = 1) or a variety of other methods (barrier, *n* = 1; coital, *n* = 1; emergency contraception and coital methods, *n* = 1; the pill, *n* = 3; LARC, *n* = 1; permanent contraception in combination with another method, *n* = 2).

Figure 2 also highlights the ways that patterns in birth control use and preferences differ based on the direction of contraceptive coercion experienced. For example, among those who experienced downward coercion (*n* = 15), almost all (*n* = 14) would prefer to use methods that require a prescription or procedure. Conversely, those who experienced upward coercion (*n* = 34) commonly reported using the birth control pill (*n* = 17) despite preferring to use other methods or not use any birth control at all.

## 4. Discussion

In this study, we examined the relationship between healthcare provider-based contraceptive coercion experienced at participants’ last contraceptive counseling and their use or non-use of their preferred contraceptive method(s). First, we found a statistically significant negative relationship between experiences of downward contraceptive coercion and the odds of using a preferred contraceptive method(s). Second, we found that the relationship between upward coercion and preferred contraceptive use was not statistically significant. Third, we highlighted patterns and nuances in current and preferred contraceptive use for those who experienced contraceptive coercion, documenting lack of access to desired permanent contraception and frequent use of the pill despite misalignment with participants’ preferences.

First, our findings suggest that patients who have perceived pressure from a healthcare provider to *not* use contraception are less likely to be using their preferred contraceptive method(s). In addition to being supported by an existing regional study [1], this connection is logical given that being denied desired contraception, by definition, makes it more difficult for a patient to access their desired contraception. Undoubtably, some patients manage to navigate and overcome this barrier, whereas others remain unable to access their desired contraception. However, little is known about this navigation process. Future research should explore who is and is not able to access desired contraception after being refused their preferred method and how they manage to do so. Furthermore, as we learn more about the manifestations and impacts of downward contraceptive coercion, it may be important to distinguish between the roles of contraceptive denials based on provider biases, misperceptions, and miscommunications versus legitimate medical contraindications.

Second, although patients who perceived pressure from a healthcare provider to use contraception were also less likely to be using their preferred contraception, this relationship was not statistically significant in this study. It is unlikely that this null finding is due to limited statistical power given that upward coercion is more common than downward coercion (experienced by 15% versus 4% of the study sample, respectively). Regardless, this null finding is surprising given the connection between upward coercion and preferred contraceptive use found in the existing research [1]. Moreover, it stands to reason that pressuring patients to use contraception, or a specific type of contraception, would make them more likely to use that method of contraception regardless of their preferences. There are several possible reasons that we did not find this to be the case. For example, it is possible that this null finding is related to the cross-sectional nature of our data. After initiating a contraceptive method, patients’ retrospective perception of the pre-use desirability of that method may shift, similar to how feelings about the perceived desirability or intendedness of a pregnancy can change in the months and years after delivery [25,26]. Importantly, regardless of how these perceptions may shift over time, care must center patients’ contraceptive preferences and priorities. This will require advocating for system-wide change, building provider capacity for patient-centered care, and working to eliminate contraceptive coercion.

Another possible reason for this null finding is that, even when they face provider-based contraceptive coercion, patients may advocate for themselves in ways that allow them to access their desired contraceptive methods (or to exercise their right to *not* use contraception if that is their desire). Building patients’ health literacy, self-efficacy, and ability to advocate for themselves in their healthcare-seeking is more important than ever in the current U.S. healthcare landscape, where legal and financial barriers are prevalent [14,27], providers are given limited time with patients [28], and patients are expected to be informed and active healthcare consumers [29].

Third, this study highlights patterns in preferred contraceptive use for people who have experienced contraceptive coercion, suggesting that provider-based coercion functions as a barrier to accessing desired permanent contraceptive methods including tubal ligation and vasectomy. Our study also suggests that patients who have experienced contraceptive coercion often use the birth control pill even when this is not their desired contraceptive method. These findings fit with emerging research that documents pressure to use the birth control pill and permanent contraception refusals as the most common manifestations of upward and downward coercion, respectively [2]. The current study adds additional nuance by documenting that, not only are these manifestations of contraceptive coercion relatively common, but they are potential mechanisms preventing patients from using their preferred contraceptive method(s) (or choosing not to use contraception).

### Limitations and Strengths

This study’s limitations include our non-probability sampling approach, which limits the representativeness of the sample and the generalizability of the findings. Notably, our sample overrepresents sexual minority individuals, which could impact the findings and their generalizability. Given that emerging evidence suggests that sexual minorities are at increased risk of contraceptive coercion [30], the rates of coercion in our sample may be higher than those in the general population. Our findings are consistent even when controlling for sexual orientation, although additional studies are needed in order to establish the national prevalence of contraceptive coercion and its relationship with contraceptive use.

Additionally, our cross-sectional data limit the conclusions we can draw about the time order of our study variables and the causal relationship between them. Furthermore, since participants reported about coercion experienced at their last contraceptive counseling visit, and these visits ranged from sometime in the past year to 7 years or more in the past, issues related to recall bias could come into play, potentially causing participants to over- or under-report coercion. We have attempted to account for these issues by including time since last visit as a control variable and running sensitivity analyses limiting the sample to only those whose last contraceptive counseling was in the past year. Regardless, future research could address these issues during data collection by assessing coercion immediately following a clinical encounter.

Finally, the sample size used to generate Figure 2 (*n* = 44) is relatively small. This represents a subsample of study participants who experienced contraceptive coercion and were not using their preferred contraceptive method(s). Despite the relatively small subsample, this analysis provides important information about a key group of people: those who experienced coercion in their contraceptive care and have unmet contraceptive needs [31]. Understanding this patient group’s needs and experiences is critical to bolstering patient-centered care and promoting reproductive autonomy. Future research with a larger sample could extend the reach of our subanalysis.

This study produces important preliminary findings, and future research using nationally representative longitudinal data would help establish the impact of contraceptive coercion on contraceptive behavior and preferred method use. Additionally, future intervention research is needed to establish clinical- and system-level solutions to prevent coercion and promote reproductive autonomy. Such efforts could include improving patient capacity for self-efficacy and self-advocacy and reforming both medical training and practice to improve patient-centeredness and respect patients’ contraceptive choices. Important strengths of this study include our diverse national sample and our patient-centered approach. Patients reported their perceptions of coercion in their contraceptive care and defined the outcome variable themselves by indicating whether they are using their preferred method(s), including an option to indicate their preference to not use any contraception.

## 5. Conclusions

Although both upward and downward contraceptive coercion are counter to patient-centered care, in this study, patients who experienced downward coercion or pressure from a healthcare provider to *not* use birth control were less likely to be using their preferred contraceptive method. Promoting reproductive autonomy requires contraceptive care that is comprehensive, patient-centered, and without bias.

## Figures and Tables

**Figure 1 healthcare-13-00145-f001:**
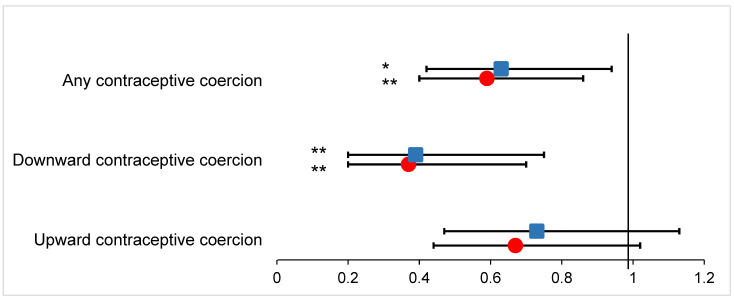
Logistic regressions of preferred contraceptive method(s) use regressed on contraceptive coercion at last contraceptive counseling among a 2023 U.S. sample of reproductive-aged people assigned female at birth who had ever talked to a healthcare provider about birth control (*N* = 1197). Figure notes: * *p* < 0.05, ** *p* < 0.01. Each point estimate and error bar represent an estimate from a different logistic regression model. Point estimates are odds ratios, and error bars are 95% confidence intervals. Red circles are unadjusted coefficients, and blue squares are adjusted coefficients. Adjusted estimates control for age, education level, health insurance status, race/ethnicity, sexual orientation, marital status, and time since last contraceptive counseling. Tables listing coefficients for the controls are included as Appendix A.

**Figure 2 healthcare-13-00145-f002:**
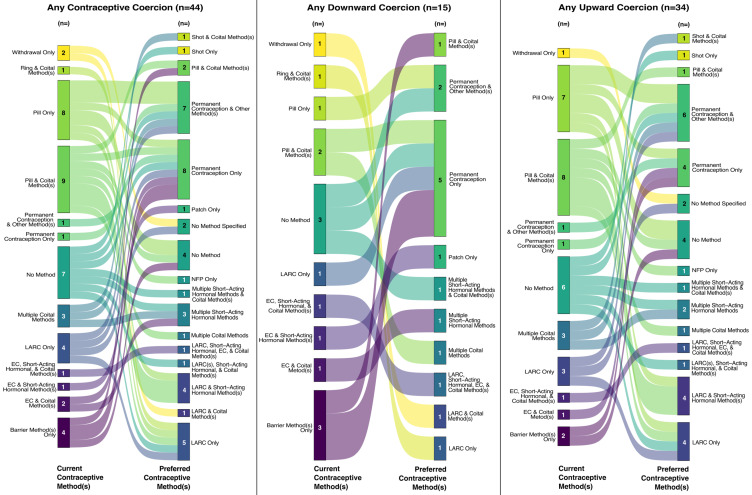
Mapping current and preferred contraceptive method(s) for the subsample of participants who experienced coercion and were not using their preferred method(s) among a 2023 U.S. sample of reproductive-aged people assigned female at birth who had ever talked to a healthcare provider about birth control. Figure notes: EC = emergency contraception; LARC = long-acting reversible contraception (includes intrauterine devices and implants); NFP = natural family planning; coital methods include condoms, diaphragm, cervical cap, sponge, contraceptive foam/jelly/cream, withdrawal, and natural family planning; short-acting hormonal methods include the birth control pill, shot, ring, and patch; permanent contraception includes vasectomy, tubal ligation, hysterectomy, and Essure.

**Table 1 healthcare-13-00145-t001:** Sample characteristics for a 2023 U.S. sample of reproductive-aged people assigned female at birth who had ever talked to a healthcare provider about birth control (*N* = 1197).

Variable	*n* (%)
Gender identity	
Cisgender woman	1137 (94.9%)
Trans man assigned female at birth	16 (1.3%)
Nonbinary assigned female at birth	44 (3.7%)
Age, Mean (Standard Deviation)	32.78 (8.08)
Education level	
High school or less	141 (11.8%)
Associate degree or some college	390 (32.6%)
Bachelor’s degree	444 (37.1%)
Graduate school	222 (18.6%)
Health insurance status	
Not insured	109 (9.1%)
Insured	1088 (90.9%)
Race/Ethnicity	
Hispanic	101 (8.4%)
Non-Hispanic Asian	83 (6.9%)
Non-Hispanic Black	182 (15.2%)
Non-Hispanic mixed race	48 (4.0%)
Non-Hispanic White	783 (65.4%)
Sexual orientation	
Bisexual	245 (20.5%)
Heterosexual	779 (65.1%)
Gay or lesbian	53 (4.4%)
Asexual, pansexual, queer, questioning, or prefer to self-describe	120 (10.0%)
Marital status	
Not married	777 (64.9%)
Married	420 (35.1%)
Time since last contraceptive counseling	
In the past year	559 (46.7%)
1–3 years	412 (34.4%)
4–6 years	136 (11.4%)
7 years or more	90 (7.5%)

**Table 2 healthcare-13-00145-t002:** Distribution of coercion at last contraceptive counseling and use of preferred contraceptive method(s) among a 2023 U.S. sample of reproductive-aged people assigned female at birth who had ever talked to a healthcare provider about birth control (*N* = 1197).

Coercion at Last Contraceptive Counseling	Total	Using Preferred Contraceptive Method(s)	Not Using Preferred Contraceptive Method(s)
	*n* (%)	*n* (%)	*n* (%)
Any coercion			
No	976 (81.5%)	851 (87.2%) **	125 (12.8%) **
Yes	221 (18.5%)	177 (80.1%) **	44 (19.9%) **
Downward coercion			
No	1146 (95.7%)	992 (86.6%) **	154 (13.4%) **
Yes	51 (4.3%)	36 (70.6%) **	15 (29.4%) **
Upward coercion			
No	1014 (84.7%)	879 (86.7%)	135 (13.3%)
Yes	183 (15.3%)	149 (81.4%)	34 (18.6%)

Note: Downward coercion = pressure to not use contraception; Upward coercion = pressure to use conracption. ** *p* < 0.01, from chi-square tests comparing frequencies.

## Data Availability

Data are not publicly available due to privacy and ethical restrictions.

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
