# Peer review of "Patient Preferences or Provider Pressure? The Relationship Between Coercive Contraceptive Care and Preferred Contraceptive Use"

_healthcare, 2025, doi:10.3390/healthcare13020145_

Round 1
Reviewer 1 Report
Comments and Suggestions for Authors
1. More details required about the patients or participants required, such as economic status, health status, problem of contraceptive choices, why they need go with providers choice for the study.
2. What are the post consequences are there after the downward coercion, list out with sample wise and age wise to describe it more accurately.
3. Is it US government made it these contraceptive choices to public, is the health providers trained properly, in contraception research so many advanced methods are there, why not applying with all the population apart from classic methods and risky methods
4. Mentioned some the comments in the discussion and conlcsuion part....work on that without fail

Reviewer 2 Report
Comments and Suggestions for Authors
Dear Authors,
Thank you for the opportunity to review your manuscript titled "Patient preferences or provider pressure? The relationship between coercive contraceptive care and preferred contraceptive use." I found your study to be an important contribution to understanding the dynamics of contraceptive counseling and patient-centered care. Below, I have outlined specific comments and suggestions for each section of the manuscript:
Abstract
Consider highlighting how your findings advance understanding or inform interventions aimed at reducing contraceptive coercion.
The keywords are relevant but could be expanded to improve discoverability. Adding terms like "healthcare provider bias" or "patient-centered healthcare" might help.
Introduction
Consider expanding on the importance of using a diverse, national sample and how it builds on previous studies conducted in more limited settings, such as the Appalachian region.
The objectives of the study are clear, but a concise statement of the study’s primary aim could help maintain focus.
Methods
I suggest elaborating on how oversampling was addressed in the analysis and its potential influence on the findings.
Discussion
1. The discussion effectively connects your findings to existing literature. However, I encourage you to delve deeper into the practical implications of your study, particularly for healthcare provider training and systemic changes in contraceptive counseling.
o Propose specific strategies to reduce coercion in contraceptive care and promote patient-centered practices.
2. While the limitations are acknowledged, their potential impact on the study’s conclusions warrants more elaboration.
o For instance, discuss how recall bias or non-probability sampling might have influenced the results.
Conclusion
1. The conclusion effectively summarizes the findings but could benefit from actionable recommendations for practitioners or policymakers.
o Consider including a brief discussion on how your findings can inform healthcare guidelines or training programs aimed at improving patient-centered contraceptive care.
2. Suggestions for future research are relevant but could be more specific. Highlighting areas such as longitudinal research or intervention studies would strengthen this section.
Overall, this manuscript addresses a critical and timely issue in reproductive healthcare. I hope these comments help you refine your work further and strengthen its impact on the field. Thank you again for your valuable contribution. I noticed that in several places within the manuscript, the period appears before the citation. I recommend reviewing the entire manuscript and ensuring that all citations are formatted consistently with this style.
Round 2
Reviewer 2 Report
Comments and Suggestions for Authors
Approved